# A Changing Home: A Cross-Sectional Study on Environmental Degradation, Resettlement and Psychological Distress in a Western German Coal-Mining Region

**DOI:** 10.3390/ijerph19127143

**Published:** 2022-06-10

**Authors:** Theresa Krüger, Thomas Kraus, Andrea Kaifie

**Affiliations:** Institute for Occupational, Social, and Environmental Medicine, Medical Faculty, RWTH Aachen University, Pauwelsstrasse 30, 52070 Aachen, Germany; theresa.krueger@rwth-aachen.de (T.K.); tkraus@ukaachen.de (T.K.)

**Keywords:** solastalgia, environmental change, place attachment, home environment, mental health, psychological stress, depression, coal mining, relocation

## Abstract

Unwelcome environmental changes can lead to psychological distress, known as “solastalgia”. In Germany, the open-pit mining of brown coal results in environmental changes as well as in the resettlement of adjacent villages. In this study, we investigated the risk of open-pit mining for solastalgia and psychological disorders (e.g., depression, generalized anxiety and somatization) in local communities. The current residents and resettlers from two German open-pit mines were surveyed concerning environmental stressors, place attachment, impacts and mental health status. In total, 620 people responded, including 181 resettlers, 114 people from villages threatened by resettlement and 325 people from non-threatened villages near an open-pit mine. All groups self-reported high levels of psychological distress, approximately ranging between 2–7.5 times above the population average. Respondents from resettlement-threatened villages showed the worst mental health status, with 52.7% indicating at least moderate somatization levels (score sum > 9), compared to 28% among resettlers. We observed a mean PHQ depression score of 7.9 (SD 5.9) for people from resettlement-threatened villages, 7.4 (SD 6.0) for people from not-threatened villages, compared to 5.0 (SD 6.5) for already resettled people (*p* < 0.001). In conclusion, the degradation and loss of the home environment caused by open-pit mining was associated with an increased prevalence of depressive, anxious and somatoform symptoms in local communities. This reveals a need for further in-depth research, targeted psychosocial support and improved policy frameworks, in favor of residents’ and resettlers’ mental health.

## 1. Introduction

### 1.1. The Concept of Solastalgia

Health and wellbeing greatly depend on the environment and on the surrounding ecosystems and landscapes on which our natural livelihoods are built. In today’s globalized world, ecological disturbance is ubiquitous, whether due to resource extraction, infrastructural projects, population growth or climate change. The idea that ecosystem integrity is vital for healthy human societies is also reflected in the recent holistic concept of planetary health [1].

The integrity of the direct home environment is a major key element for psychosocial health, contributing to our identity, security, culture and sense of belonging [2,3]. The distress caused by the transformation and disruption of familiar places, such as peoples’ homes, is condensed under the concept of “solastalgia”. It describes a potential human reaction when valued physical and social environments are negatively transformed and deprived of their capacities to give solace [4,5]. A central aspect of solastalgia that distinguishes it from related concepts on ecosystems and human health relationships, such as eco-anxiety or ecological grief [6], is its explicit focus on place: solastalgia is a “place-based lived experience” [7].

The term was introduced by the environmental philosopher Glenn Albrecht after having undertaken fieldwork in an open-pit coal mining area in western Australia [8,9]. It has since found growing empirical application in the contexts of resource extraction [10,11,12,13], natural disasters [5,14] and climate change [15,16,17], showing that both acute and chronic factors can cause solastalgic distress.

It is well documented that the ecologic disturbance of the home environment can result in solastalgic feelings, such as grief, desolation, loss of identity and powerlessness [2,9,18]. However, as yet it is unclear to what extent these feelings can escalate to serious mental disorders, such as depression, or to generalized anxiety disorders [19]. The experience of solastalgia is often linked with diminished or disturbed contact to natural environments [5,12,14]. There is growing evidence on how contact with nature or green spaces is tied to better long-term health outcomes, even after controlling for socioeconomic factors. These outcomes include mental and cognitive conditions such as depression, anxiety and attention deficit hyperactivity disorder, as well as physical conditions, such as obesity, respiratory and cardiovascular diseases [20,21]. It can, therefore, be assumed that individuals experiencing solastalgia are also more likely to suffer from other (non-psychosomatic) health complaints (e.g., respiratory diseases, due to the air pollution in coal-mining areas [8,22]). The presence of these health complaints (or even solely the fear of ill health), in turn, could trigger/intensify psychological distress [8,23,24].

Overall, solastalgia seems to be a promising approach to highlight the interdependence of environmental integrity, place attachment and human health, which has also been recently acknowledged by the Lancet Commission on Health and Climate Change [21], the American Psychological Association [2] and the World Health Organization [25].

At a UN level, combating solastalgia in the context of fossil fuel extraction can also contribute to the 2030 agenda, most notably to the following Sustainable Development Goals: 3 (Good Health and Well-being), 7 (Clean Energy), 13 (Climate Action) and 15 (Life on Land) [26].

### 1.2. Mining-Induced Environmental Degradation and Relocation

Most industries in the world depend on minerals and mineral products, and for this reason mining is carried out in nearly every country [27], often leaving behind a range of environmental and social disturbances [28]. The scope, scale and systemic nature of resource extraction (e.g., mining) makes it difficult to comprehensively assess its overall impacts on nature, human health and societies [29].

Wherever large-scale mining takes place, it may become necessary to remove landscapes, farmland, infrastructural facilities or even entire villages. Development projects are assumed to be the second largest cause of resettlement, after disasters, displacing around fifteen million people annually [30]. At least one in ten cases of development-induced displacement and resettlement (DIDR) worldwide is due to mining, this is referred to as mining-induced displacement and resettlement (MIDR). Although it is often presented as a problem specific to low- and middle-income countries, MIDR is actually a global phenomenon that also occurs on the European continent [31].

Given its impactful and often irreversible nature, mining can be expected to disrupt both ecological and social environments, with possible consequences for health and life quality. While there is a growing body of research on health and resource extraction, Brisbois et al. [32] described the impacts on mental health and wellbeing as still being “neglected topics”. Moreover, a low research priority has been placed on affected communities, compared to the workers [32]. In the contexts of DIDR, MIDR and mental health, there is only a small amount of research [33], which has focused predominantly on low- and middle-income countries [34,35]. Our study addresses these research gaps, exploring the mental health impacts of environmental degradation and MIDR in western Germany.

### 1.3. Lignite Mining in Germany

In 2018, Germany was the largest producer of lignite (also known as soft or brown coal) in the world, which has played an important role in shaping the country’s economic and social structure for decades. It is used almost entirely for domestic power generation [36]. Currently, lignite is mined in the form of open pits in the following three German coal fields: the Rhenish (Rhineland, West Germany), Lusatian (Lusatia, East Germany) and central German coal fields [37]. So far, lignite mining has contributed to the relocation of more than 300 villages [38] in the country, totaling approximately 120,000 inhabitants [39].

### 1.4. Study Aim

While studies suggest that both local environmental degradation and forced relocation can contribute to an increase in mental health problems, such as depressive disorders [10,13,35,40], the empirical evidence specific to open-pit mining in Germany has been largely missing. Furthermore, there is a general lack of quantitative data in the research field of solastalgia [7]. Therefore, the present study aimed to gather and analyze primary quantitative data, documenting the possible environmental and resettlement distress due to the expansion of German open-pit mines. We characterized and compared three groups of participants, who differed in their type of affectedness due to their residential situation (e.g., the environmental degradation in pit-edge villages, the additional threat of resettlement in old villages and the experienced resettlement in new villages). We investigated to what extent environmental change and forced relocation might be risk factors for psychological distress—namely depressive, anxious and somatic symptoms—and whether the concept of solastalgia is applicable to the mining-affected communities in western Germany.

## 2. Materials and Methods

### 2.1. Study Context and Area

A cross-sectional study was conducted between June and July 2021 in the Rhenish region. The densely populated and intensively industrialized Rhineland is Germany’s most important lignite region, covering an area of around 250,000 hectares west of Cologne. Today, three large-scale open-pit mines in the Rhenish region—named Hambach, Garzweiler II and Inden—are still operated [41]. Lignite mining in the Rhineland started in the 19th century with less large-scale, open-pit mines. After the Second World War, in the 1950s, industrial growth in the area led to an expansion of open-pit mines; however, fewer families depended on jobs in the lignite industry [42] and the acceptance of mining among the local population dropped noticeably in this time. Many residents were no longer willing to accept the pollution and loss of the highly fertile farmlands, forests, rivers and their homes [42]. With the emergence of the environmental movement in Germany in the 1980s, this opposition encompassed broader sections of the population [36]. In recent years, with the growth in population-wide climate change awareness, the conflicts in the coal mining regions have gained new dimensions and stakeholders [43,44].

Since 1950, about 130 villages, with 40,000 inhabitants, have been resettled in the Rhenish region [45]. The whole resettlement process of a community from an old village to a newly built village (“joint resettlement”) is supposed to take approximately 15 years, with 5 years of planning and 10 years of implementation [46]. This long period of time and the involvement of communities in the resettlement planning and process, as intended by the responsible authorities, allows for a certain degree of predictability and control. Willms [44] describes the relocation process in more detail and explores the reasons for both pre-emptive resettlement and prolonged delay. However, it can be assumed that the socio-economic, time, physical and mental efforts of relocation—which can involve protracted sales negotiations, the self-organized construction of a new house and leaving behind familiar structures, places and people—are causing considerable distress.

Although the German Bundestag (parliament) decided in 2020 to phase-down lignite by 2038 at the latest [47], six still partly-inhabited villages in the Rhineland remained at risk of relocation for the expansion of existing open-pit mines (at the time of our data collection, mid-2021). Other villages could also soon find themselves in closer proximity to the approaching pit edge [48].

### 2.2. Participants and Data Collection

The study population included residents of the two active open-pit mines, Garzweiler II (further referred to as Garzweiler) and Hambach, in the Rhenish region. Study participants had to be at least 18 years old and were required to live, or to have recently lived (prior to resettlement), in the immediate vicinity of the Garzweiler or Hambach open-pit mine (<7 km beeline, considered as a high disturbance area). This included residents from pit-edge villages (not threatened with resettlement), resettlement-threatened villages (further referred to as old villages) and new settlements (further referred to as new villages see Figure 1). Since 2016, five villages at the Garzweiler open-pit mine [49] and, since 2012, one village at the Hambach open-pit mine [50] have been resettled to make way for mine developments. The most recent population census of the relevant villages can be found in Table 1 but this is subject to constant fluctuation due to resettlements.

For data collection, both online and paper-based versions of the same questionnaire were used in order to reach a large number of the study population, regardless of their age and media use. The online questionnaire was generated using SoSci Survey [51] and was made available to users via the following site: www.soscisurvey.de. Participants were recruited via mailing lists and private chat groups (e.g., Facebook and WhatsApp) for village communities and associations, as well as via local newspapers, with support from the local key stakeholders, who shared the study information and access link to the online questionnaire. Self-administered paper-based questionnaires were distributed by the drop-off method to all the households in the five old villages and the corresponding five new villages, as well as in the pit-edge village of Wanlo, at the Garzweiler open-pit mine, in June 2021. Additionally, paper-based questionnaires were displayed in publicly accessible places in the pit-edge villages of Kaulhausen and Venrath (bakery) and Holzweiler (gas station). At the Hambach open-pit mine no active recruitment took place. The questionnaires were mailed back anonymously via prepaid envelopes by the end of July.

### 2.3. Instrument and Development

The questionnaire contained 87 similar items for all respondents and an additional number of items that differed in quantity and content according to the respective residential situations (21 additional items for people in new villages, 20 for people in old villages and 5 for people in pit-edge villages). The total 133 items included questions and statements about the following topics:Sociodemographics: age, gender, education, marital status, (grand)children and residential situation (e.g., ownership of property, family heritage and ancestry).Environmental hazards: dust, noise, vibration, nocturnal illumination, traffic (five-point Likert scale: nearly always–never) and mining damages (yes/no).Place attachment: emotional connection, responsibility for people, desire or duty to preserve the place (five-point Likert scale: strongly agree–strongly disagree).Feelings about the changes and solastalgia: general attitude to mining, life satisfaction, economic benefits, fear of illnesses, loss of flora and fauna, building damages, social divisions and powerlessness (five-point Likert scale: strongly agree–strongly disagree).Resettlement process and distress: perceived social, financial, mental and physical impacts (five-point Likert scale: strongly agree–strongly disagree; yes/no).Activities in response to mining or resettlement (yes/no).Patient health questionnaire (PHQ-SADS): symptoms of generalized anxiety, depression and somatization.

We extracted suitable items from the Environmental Distress Scale (EDS) and translated them into German. The suitability assessment of items was based on the literature research into local conditions, site visits and interviews with residents (including pilot-testing, see below). The EDS was developed and validated by Higginbotham et al. after having undertaken qualitative fieldwork in an open-pit coal mining area in New South Wales, to monitor the “bio-psycho-social cost of ecosystem disturbance” [52] and has since been applied in further studies [3,5,14]. The original EDS contains a subscale measuring solastalgia with nine Likert-type items and explores the feelings of grief, concern, longing and belonging in the context of damage to a valued environment. We used only six slightly modified items due to the different local conditions in western Germany. We converted each item into a numeric score, with responses reflecting the highest level of solastalgic distress coded as five and responses reflecting the lowest level of solastalgic distress coded as one (i.e., strongly agree = 5 and strongly disagree = 1), and only considered questionnaires with all six items answered.

The PHQ-SADS was used to detect the levels of psychological distress. Its scales are well-validated for the detection and monitoring of depression, anxiety and somatization, and they are broadly used within Germany [53,54,55,56] and internationally [57], they were also already partly employed in a solastalgia context [13]. The PHQ-SADS contains a 15-item scale for somatic symptoms (PHQ-15), with a total score ranging from 0 to 30, a 7-item scale for generalized anxiety (GAD-7), with a total score ranging from 0 to 21, and a 9-item scale for depressive disorders (PHQ-9), with a total score ranging from 0 to 27 [57]. Missing responses were coded with 0, assuming that the respective symptom did not occur or apply. Scores of ≥5, ≥10 and ≥15 represent mild, moderate and severe levels of somatization, generalized anxiety or depressive disorders, respectively. We classified the participants into two groups, depending on whether they had at most mild (score 0–9) or at least moderate symptom severity (≥10), as this is common practice in most diagnostic analyses with this screening tool [57].

**Table 1 ijerph-19-07143-t001:** Overview of Study Participants and Population Levels at Garzweiler Open-Pit Mine.

	Study Participants	Population Levels ^#^
	*n* (% of Overall Population)	*n*
**pit edge villages** *	322	
Wanlo	195 (18)	1087
Kaulhausen/Venrath	76 (7)	1146
Holzweiler	48 (3)	1400
others	3	
**old villages**	112 (21)	540
(Keyenberg, Westrich **, Kuckum, Berverath)		
**new villages** ***	173 (30)	574
(named like old villages with appendix ‘neu’)		

* respondents from pit edge villages were asked to specify their village to ensure they meet inclusion criteria; ** occasionally referred to as two villages (Unterwestrich and Oberwestrich); *** respondents who did not participate in the “joint settlement” to new-built villages but moved elsewhere (11.2%) included; ^#^ data from 30 June 2021 [58], for Wanlo from 31 December 2021 [59].

Further questions regarding the situation in the Rhenish region were added based on the initial site visits and interviews with residents from all three study groups. For most of the questions regarding sociodemographics, place attachment, the felt impacts of change and resettlement, a dichotomization of the Likert scale was performed for the analyses. Generally, two answers (e.g., strongly agree and agree) were combined and compared with the other three (e.g., neither agree nor disagree, disagree and strongly disagree).

Questionnaires were pilot-tested by five key stakeholders (residents) onsite, who provided written and oral feedback to improve comprehensibility, suitability and sequence of items.

### 2.4. Data Analysis

Survey data were analyzed with SAS Software (SAS 9.4, SAS Institute Inc., Cary, NC, USA). After exclusion criteria were applied (*n* = 29 entries deleted due to no information on residential status, *n* = 19 entries deleted due to residence outside the defined mining-affected area), questionnaires from a total of *n* = 620 participants were included in the analyses (*n* = 208 online, *n* = 412 paper-based). Descriptive analyses were performed for all respondents and categorized as follows: by belonging to the group of new villages, old villages or pit-edge villages. Chi-Square tests and Kruskal–Wallis H tests were used to describe the distribution of categorical and continuous variables between three study groups. Non-parametric tests were used based on the assumption of non-normal data distribution. Frequency tables were created for each variable in the Likert scales and the dichotomous questions, and the overall trends were examined. For associations between the PHQ scores, solastalgia and the period of resettlement, the Pearson correlation coefficient was used. All statistical tests were two-sided, and *p* < 0.05 was used as the level of significance.

### 2.5. Ethical Considerations

Ethical approval for the study was obtained from the local Human Research Ethics Committee of the RWTH Aachen University Faculty of Medicine (EK104-21, 8 April 2021). All participants gave written informed consent prior to filling out the questionnaire. Information about where to seek psychological support should participation in the survey cause distress was provided to respondents.

## 3. Results

### 3.1. Sample Overview

A total of 620 respondents were included in the analyses. Different numbers of questionnaires were returned from each study group, as follows: pit-edge villages (*n* = 325), new villages (*n* = 181) and old villages (*n* = 114). A large majority of 607 respondents (97.9%) originated from the Garzweiler open-pit mine. A more detailed grouping by village can be seen in Table 1. People using the online questionnaire were younger (49.6 vs. 57.2 years) and more often from pit-edge villages (64.4% vs. 46.4%—data not shown). The period since the completed resettlement for new villages’ residents was on average 32.8 months (45.1 SD), with a median of 22 months (*n* = 175).

### 3.2. Sociodemographics

Table 2 gives respondents’ sociodemographic characteristics (for exact test values see Appendix A). The average age of respondents was higher than that of the overall local population (approximately 46.8 years in Wanlo [60] and 45.1 years in Erkelenz, which includes all other new, old and-pit edge villages [61]). Moreover, the proportion of female participants was higher than in the overall local population (44.3% females in old villages, 50.3% females in new villages and approximately 48.9% to 51.4% in pit-edge villages [58,62]). Respondents from the three groups were similar concerning age, gender, university degree, property ownership, having children or grandchildren in the village and whether they had spent their entire life in this place. However, participants from old villages were less often married or in a partnership (67.6% vs. 86.7% in new and 81.1% in pit-edge villages: *p* < 0.001), had a longer ancestry (more generations) in the region (73.6% vs. 62% in new and 60.5% in pit-edge villages: *p* < 0.05) and rather lived on old family property (63.7% vs. 50% in new and 41.7% in pit-edge villages: *p* < 0.001).

### 3.3. Solastalgia and the Patient Health Questionnaire

The solastalgia scores (shown in Table 2, for exact test values see Appendix A) differed significantly among the three groups (*p* < 0.001), with people still living at the open-pit mine in either old villages (25.59, 5.81 SD) or pit-edge villages (25.38, 4.97 SD) scoring higher than those already resettled to new villages (21.19, 7.51 SD). These effects remained when the data were categorized into male and female (*p* < 0.001). Within all three groups, female respondents showed higher solastalgia levels than male respondents, however, this difference was only marginal for old villages.

The PHQ scores (shown in Table 2, for exact test values see Appendix A) for the three scales of somatic, anxious and depressive symptoms also showed considerable differences between the three groups (*p* < 0.001), even when categorized by gender (*p* < 0.001 to *p* < 0.05). Scores for all three PHQ scales were the highest in the old villages, followed by the pit-edge villages and lastly the new villages. This same trend was also found after dichotomization, based on the proportion of respondents with a moderate-to-severe symptom level: at least moderate levels were stated for somatic symptoms by 52.7% of respondents in old villages, by 46.5% in pit-edge villages and by 28% in new villages; for generalized anxiety symptoms by 45.4%, 31.2% and 18.6%, respectively; and for depressive symptoms by 34.3%, 30.3% and 20.8%, respectively. In terms of gender, specifically, this effect was only absent (i.e., not significant) for males with at least moderate anxiety or depression. Furthermore, the proportion for females with at least moderate depressive symptoms was slightly higher in pit-edge villages than in old villages. Within the three groups, the female respondents presented higher symptom severity than the male respondents; however these differences were less prominent in the new villages. Thus, people who still lived at the open-pit mines showed higher degrees of mental health problems—namely somatic, depressive and generalized anxiety symptoms—than those who had already resettled.

We detected moderate positive correlations between solastalgia and symptoms of somatization (r = 0.54 to 0.42), generalized anxiety (r = 0.51 to 0.38) and depression (r = 0.53 to 0.35), respectively (all *p* < 0.001, see Table 2). Furthermore, the three PHQ symptom scales were highly intercorrelating in the three groups (r = 0.92 to 0.72, see Appendix A), indicating comorbidity. For the people in new villages, no significant correlations between the period since they completed their resettlement and their solastalgia levels, or their somatic, anxious or depressive symptoms were found (r = −0.02 to 0.07, see Appendix A).

### 3.4. Environmental Hazards

The frequencies of observed environmental hazards in both the groups still living at the open-pit mines were comparatively similar (shown in Figure 2 and Figure 3). Dust was the most frequently observed hazard (73% of participants in old villages and 82.5% in pit-edge villages observed it often to nearly always), followed by noise from the open-pit mine (65.1% and 52.8%) and increased traffic (56.4% and 61.8%). Regarding vibrations from the open-pit mines (61.2% in old villages and 43.7% in pit-edge villages observed it sometimes to nearly always) and noise from the resettlement activities (71.2% in old villages and 87.7% in pit-edge villages observed it rarely or never), the results varied more between the groups, with the latter being, in general, seldomly observed. Moreover, *n* = 78 people in the old villages (70.9%) and *n* = 137 in the pit-edge villages (44.5%) said that they have experienced mining damages to their house or property (data not shown).

### 3.5. Place Attachment

Place attachment (see Table 3) was measured in terms of the respondents’ connection to the place, their responsibility for the people there and the sense of duty they felt to preserve it; the people in the new villages were asked to generally refer to their old place of residence at the open-pit mine. While the majority of people who had not been relocated (yet) still felt deeply connected to their place of residence (73.8% in old and 74.5% in pit-edge villages), this feeling was considerably lower among those who had already been relocated (39.8%). The sense of duty that was felt by people to preserve their (former) place of residence for future generations was higher in the pit-edge villages (79.1%), compared to the old (56.6%) and new villages (16.5%). It is noteworthy that the responsibility felt by people for the people in their (former) place of residence was relatively low in the two sites affected by resettlement (23.7% in new and 39.8% in old villages), compared to the non-affected pit-edge villages (79.1%).

### 3.6. Feelings about Changes

The felt impacts of the changes caused by the open-pit mines are presented in Table 3. In general, it can be observed that the people’s understanding for the expansion of the open-pit mines was low in all three groups, however, in comparison, this was more than twice as high in the new villages (26.3%) compared to the old (12.5%) and pit-edge villages (8.1%). In the new villages, 39.5% agreed that the economic benefits are important for the region, and about one in two (50.9%) considered that the funding of community projects was helpful, while the latter was only agreed to by around one fifth in the old villages (21.1%) and the pit-edge villages (17.7%). The open-pit mines also seemed to have the least impact on life satisfaction in the new villages, since only 29.5% reported that due to mining they could not enjoy life as much as they would have liked to, compared to 70.5% in the old villages with the highest impairment. While satisfaction with the authorities in monitoring the environmental impacts was generally low, people in the new villages reported the highest satisfaction (27.5% vs. 9.2% in the old and 7.6% in the pit-edge villages).

However, 38% of the respondents from the new villages were still concerned about mining threatening their health, while this was the case for 71.8% in the old villages and 75.7% in the pit-edge villages. The majority of respondents in the new villages felt powerless against the changes to their homeland (60%) and upset about the destruction of natural (65.9%) and historical sites (61%); however, these feelings were even more prominent in the two groups still living at the open-pit mine (8–9 out of 10 people agreed to feeling this way). Interestingly, all respondents indicated that they were more disturbed that future generations are not being made a higher priority (48.6% in the new villages, 80.2% in the old villages and 83.7% in the pit-edge villages) than they felt a personal duty to preserve their place for them (see Section 3.5). Disagreements over mining that divided the community seemed to be most common in the two groups affected by resettlement (58.4% in new villages, 68.5% in old villages). Intra-family conflicts were indicated less often, with the highest frequency of this in the old villages (18.4%).

The respondents in the pit-edge villages group were asked additional questions about how their living situation was affected by open-pit mining and whether they would leave their village if they could (data not shown). While 72% stated that their living situation was negatively affected, only 28% stated that they would leave the place.

### 3.7. Resettlement Impacts

Regarding the experienced or expected impacts of resettlement, the two affected groups (new villages and old villages) differed in all of the queried aspects (*p* < 0.001 to 0.05: see Table 4; for exact test values see Appendix A), with more negative perceptions generally expressed in the old villages. About one third of the respondents from the new villages felt physically (36.3%) or psychologically (33%) exhausted by the relocation process, while those numbers where even higher for those in the old villages (59.3% and 69.7%), especially with regard to mental exhaustion. Only 8.3% of the people in the old villages felt well advised by the authorities in the resettlement process, compared to 32.2% in the new villages. Notably, 53.8% of the respondents from the new villages stated that their general living conditions had improved after relocation. Surprisingly, only 7% in the new locations indicated that their professional situation had worsened, while in the old locations 30.4% assumed this to happen. Moreover, 71% of people in the old villages anticipated an additional financial burden, which exceeded the proportion of those already resettled who actually experienced this additional financial burden (42.7%). It is noteworthy that *n* = 30 respondents in the old villages had livestock, and only one of them affirmed that the livestock could be kept in equally good circumstances in their new place of residence. In the larger group of new villages, only *n* = 14 people reported having livestock, while four of them (28.6%) said that the holding conditions remained just as good as they were in their former place of residence.

### 3.8. Activities

The reported activity levels (shown in Figure 4) were the highest in the old villages, followed by the pit-edge villages and then the new villages. For example, more than half of the people in the old (57%) and pit-edge villages (53%) supported citizens’ initiatives against open-pit mining, while only one in five resettled persons (21%) stated that they did so. Nevertheless, every third person in the new villages (33%) had taken part in a demonstration against open-pit mining at least once. The rate of people attending village community meetings to discuss the open-pit mining impacts was high in all the groups (71% old, 67% pit-edge and 61% new villages).

### 3.9. Comments of the Participants

Almost one third of the participants added comments at the end of the questionnaire, providing details about their personal situation, further themes of distress/relief or feedback on the survey. The following aspects, which were not explicitly covered by the questionnaire, were often mentioned:-Uncertainty about the future was a particularly important issue in all three groups. In the old villages, respondents were uncertain about whether their villages would be finally excavated or could be preserved, in the light of recent or expected political developments. In the new villages, a few participants expressed a desire to preserve their old villages, while many described the thought of strangers reoccupying their former homes as an enormous emotional burden, knowing that their resettlement would then have been unnecessary.-The feeling of being subject to environmental and political injustice was raised frequently and was often paired with frustration due to the lack of satisfactory advisory and support services.-Insecurity due to burglaries in abandoned neighboring houses was an important concern in the old villages, accompanied by perceived disturbances due to the presence of security forces, activists, the press and curious onlookers (referred to as “ghost village tourists”).-The perceived benefits of relocation (e.g., more age-appropriate and refurbished homes and closer proximity to the city), as well as the disadvantages (e.g., living on a large construction site for years, limited recreational activities and less access to recreational spaces) were described in more detail.-Moreover, concerns about environmental degradation and climate change and the perception that they are being fueled at one’s doorstep were raised.

## 4. Discussion

This cross-sectional study aimed to record the psychological distress and the solastalgic feelings associated with open-pit mining in western Germany. We found high levels of self-reported depressive, anxious and somatic symptoms in all three groups examined (old villages, new villages and pit-edge villages), particularly in female respondents, and with participants from the old villages being the most impaired. To our knowledge, this is the first study to quantitatively assess the links between the psychological distress and the environmental degradation caused by open-pit mining in Germany.

Respondents living at the open-pit mines reported high levels of depressive, anxious and somatoform symptoms, with participants from the old villages scoring slightly higher, compared to the respondents from the pit-edge villages. These findings indicate that the gradual loss of social and community structures and one’s own home and the reconstruction of a new home can create an additional mental burden. However, people in the new villages—who are no longer directly exposed to the open-pit mine or involved in resettlement activities—still showed elevated symptom levels when comparing them with the general population norms. In the most recent German population-representative studies that have used the PHQ as a screening tool, the prevalence of at least moderate symptom severity was 5.6%, or 8.1% for depression [53,54], 5.9% for generalized anxiety [55] and 14.9% for somatization [56]. In this study, we found remarkable levels—approximately twice to 7.5 times as high as the general population norms in the open-pit mining-affected communities (see Table 2). However, it can be assumed that up-to-date general population norms for depressive disorders might be higher, thus, the discrepancies with our respondents would be lower, especially since German health insurance funds have reported increases in diagnosed cases of depression in the past years [63]. However, there is some debate about whether this trend is attributable to overall prevalence or to other factors, such as coding practice or patients’ help-seeking behavior [63,64]. Furthermore, the COVID-19 pandemic may have contributed to a generally higher mental health burden nowadays [65,66,67]. Given that our data collection took place in June to July 2021, with very low regional incidences [68] and few restrictions, we consider the impacts of COVID-19 on our results rather limited.

The fact that female participants reported higher depressive, anxious or somatic symptoms in the PHQ is a well-known phenomenon seen in previous research in the general population [54,55,56,69,70] as well as in the context of experienced environmental disturbances [5,13,71]. Regarding solastalgia research, gender is so far considered an understudied aspect [7]. Nevertheless, Elser et al. [12] observed higher solastalgia scores in female individuals, congruent with our findings.

The levels of depressive, anxious and somatic symptoms should not be equated with the prevalence of illness and do not serve as a substitute for accurate diagnostic interviews, carried out by qualified professionals. Though scores of moderate symptom severity have shown good sensitivity and specificity in the diagnosis of major depression [57], more recent studies have indicated that self-report screening questionnaires overestimate the prevalence of both depression and generalized anxiety, compared with diagnostic interviews [72,73]. However, given that special barriers to seeking help and care for people with psychological problems exist in rural communities, such as a culture of self-reliance and a lack of anonymity [74,75], our results remain alarming.

Based on the questionnaire, including the additional comments of the respondents, we identified the following risk factors for psychological distress (without weighing their importance, or claiming exhaustiveness or a direct causal relationship): environmental hazards, such as dust and noise from open-pit mining operations and resettlement works; fear of ill health from those and other hazards or pollutants; solastalgia, due to unwelcomed environmental change; loss of familiar places (e.g., home, land and property); negotiation and relocation-related stress and workload; community and family divisions and erosions; future uncertainty; felt powerlessness and environmental injustice; disturbances from activism, the press and curious members of the public; nostalgia; and uprooting. Similar themes of mental distress have been reported by residents of open-pit mines in Australia, for example, regarding personal health; damages to homes, properties, landscapes and community heritage; higher costs of living; changing neighborhood structures; and social pressure caused by mining companies and the mistrust between the supporters and opponents of mining [8].

While individual experiences, emotions and reactions are manifold, we observed a trend reflecting the high psychological distress in open-pit mining communities, whether affected by resettlement or not. The mental burden appears to be(come) lower for those that have become distanced from the open-pit mine. This observation could be attributed to two different mechanisms. First, it could be based on exposure, ergo moving away from the open-pit mine would provide relief. Second, it could be due to the different characteristics of the two groups, also shedding light on the rationales for early resettlement or prolonged stay.

Despite differences between the groups, it is conceivable that a greater day-to-day exposure to open-pit mining and its impacts is the most important driver of psychological distress and solastalgia. These assumptions align with prior studies that describe the adverse effects on the mental health and wellbeing of local communities exposed to open-pit coal mining [10,13] and other industrial projects, such as oil and gas extraction sites [71,76,77], petroleum refineries [78] and waste dumps [79]. For instance, Hendryx et al. [13] found that residing in an area where mountaintop-removal coal mining is practiced, poses a relative risk for mild and moderate (but not severe) depression, using the PHQ-8 (PHQ-9 with one item less) as a screening tool. A similar conclusion was reached in the study by Canu et al. [10], where residents of coal-mining counties had approximately 37% higher odds of being diagnosed with a depressive disorder, compared to those in non-mining counties, based on an emergency department database. In contrast to our research, no increase in the risk of anxiety disorders was found here. Furthermore, it is an inherent feature of the concept of solastalgia, that solastalgic feelings are strongest when people immediately experience the unwelcome change in their homeland [52]. Thus, solastalgia diminishes when one no longer witnesses how the valued place is negatively transformed, which is consistent with our findings.

It is noteworthy that perceiving the open-pit mine as a health threat—which is what approximately three quarters of the people living in its vicinity do and which was very often echoed in the comments—can result in emotional reactions, such as fear or anxiety. According to the five-stage stress-coping model from Higginbotham et al. [52], the further threat appraisal can lead to action- or emotion-based copying and, finally, to adapting. Importantly, this threat appraisal is iterative, so the threat is constantly reassessed, and responses vary according to personal situations and resources [78], which may also explain the decreased threat appraisal of resettlers (i.e., moving away from the open-pit mine as a form of action-based copying). Importantly, the subjective threat appraisal is paramount in generating emotional or psychological distress, rather than a real health risk, which is why nocebo effects may occur, ergo the expectation of illness from mining could already trigger (mental) illness [24]. In addition, mental disorders such as depression or anxiety can also increase the risk of developing non-psychiatric conditions, e.g., obesity [80], which, again, highlights the various impacts and comorbidities of solastalgia.

Notably, though the place of residence was the key distinguishing criterion among the three studied groups, further observed differences may be relevant. Participants from the old villages seemed more anchored to their homeland, as measured by longer family roots in the region and by a higher rate of residens living on old family property. A congruent observation was made in a coal mining community in south-eastern Australia, where environmental distress, including solastalgia, was related to having a long family heritage in the area and occupying a heritage family home [52]. Deeper familial embeddedness may also be one reason why people in old villages apparently considered relocation more difficult, as evidenced by their not having completed it yet or by not even having started negotiations. In line with this, more participants in the old villages (exactly one in two) have spent their entire lives in their village, compared with respondents from the new villages, however, these differences were not significant. Moreover, participants from the old villages were significantly less likely to be married or in a partnership, which was identified as a sociodemographic correlate for depression in previous research that was conducted concerning both coal-mining-affected communities and the general (German) population [13,81].

Also, a remarkably high rate of participants from old villages claimed that they have livestock. It is recognized that compensating residents with livestock, and, therefore, often larger plots, is more challenging for the mining company, since available land is scarce [82].

Moreover, delaying resettlement can be considered as an “act of resistance against the normality of displacement” [44], likely to be “committed” by those who have a more negative perception of the open-pit mine and the mining company, who feel predominantly upset or disturbed about the destruction of their homeland, nature, buildings and future generations’ perspectives and are more often engaged in activities against mining. Community resistance to mining projects is known to be more likely to occur when the experienced environmental impacts are large and the level of participation and trust towards responsible institutions are low [83]. Individual or collective activities could be coping mechanisms that strengthen self-efficacy and, thereby, undermine perceived powerlessness [84,85]. However, it may also be that these activities place an additional time, physical and mental burden on the respondents who are involved and, thus, contribute to higher PHQ scores [84,86].

Eventually, it is conceivable that resettling may improve community mental health by less exposure to the open-pit mine and its impacts. Conversely, keeping in mind the empirical factors of vulnerability, relocating residents from old villages may also result in a persistence of symptoms or in further psychosocial impairment. The cross-sectional design of this study does not allow us to draw definitive conclusions on this. Given the fact that approximately half of the respondents from the new villages stated that their living conditions had improved, we suggest that the overall resettlement outcome varies widely. This is further reflected by the fact that most of the resettlement-affected participants (in the old and new villages) reported divisive disagreements over the open-pit mine within their communities.

While it is possible for the respondents in the old villages that resettlement distress contributes more to mental health problems than the mourning for environmental degradation (solastalgia) itself, the findings from pit-edge villages suggest otherwise. Though the demolition of the neighboring villages could also have a secondary impact on pit-edge villages, the approaching open-pit mine and the resulting environmental impacts seemed to be the single biggest source of distress for them.

### 4.1. Limitations

Several limitations of this study should be taken into consideration, the first and foremost of which is its cross-sectional design, which does not allow for causal inference about the role of open-pit mining on the emergence or exacerbation of psychological disorders. A lack of pre-resettlement data also makes relocated residents vulnerable to recall and selection bias.

Second, caution is advised when applying our results to the entire community. As indicated in Section 3.2, our study participants were older and more often female than the overall local population. The age discrepancy may be due to the inclusion criteria (no participation under 18 years). The different numbers of returned questionnaires from our three groups may be caused by varying overall population levels (see Table 1); however, since we could not test for non-response bias, more “environmentally aware” residents or those disturbed by environmental hazards, mining-induced changes or resettlement, may have completed the long survey. Although we were presumably able to make the questionnaire available to all households in the old and newly-built villages at the Garzweiler open-pit mine by using the drop-off method, we reached only a small number of those resettlers who did not participate in the “joint resettlement” but moved elsewhere (applies to approx. 40% of the resettlers in general) [49]. Similarly, the views of underage and non-German-speaking individuals are not reflected in this study. Therefore, representativeness remains limited.

Third, our questionnaire contained more statements postulating negative than positive feelings about the environmental changes and resettlement, hence, there could be an acquiescence bias.

Fourth, no exact response rate can be given due to the explorative study design, with a combination of online and paper-based participation, using the drop-off method, as well as a public display of the questionnaires. However, an imprecise equivalent for the response rate can be found in Table 1 (proportions of participants in the overall village populations).

Fifth, there was a lack of an appropriate control group in this study; an appropriate control group would have been one that did not have a local coal mining background but that had comparable regional and socio-cultural characteristics. Thus, the observed mining-specific risk factors for mental health remain preliminary.

Lastly, some relevant themes may have been missed in our regular questionnaire, as can be assumed by comments made by the participants (see Section 3.9).

### 4.2. Implications and Future Directions

Due to a general lack of data on environmental distress and solastalgia and the great interest that the study has attracted in the local population, further in-depth research appears indispensable. Future studies should employ longitudinal approaches to assess the long-term psychosocial consequences of relocation, as well as those of further mining developments, allowing conclusions about cause–effect relationships. The aforementioned missing themes (see Section 3.9) could also be incorporated into future questionnaires, or further explored through mixed-method research approaches. This could further include, for example, existing psychological diagnoses or substance use, which are known to be more frequent in mining than in non-mining communities [9], as well as other (mental) health conditions that are linked with resource extraction or solastalgia. Information about the precise progress of the resettlement process and employment dependencies with the mining company or related corporates (making residents less likely to criticize a project [44]) could further contextualize the study’s findings. The application of standardized psychometric place attachment scales (e.g., APAS [87]) could shed more light on place characteristics and allow a comparison of the findings across communities. A survey by age groups (with a sufficient sample size) would also be informative, as it seems likely that the stresses and strains vary by age (e.g., given physical decline in later life, resettlement and familiarization with the new village could be disproportionally difficult). Moreover, there is a pressing need for more studies focusing on youths, who are, in general, underrepresented in solastalgia research [7], even despite the evidence suggesting that climate change puts a disproportionate psychological burden on the younger generations [88].

Our findings indicate a need for psychosocial support. Low-threshold offerings from open or outreach counseling to psychotherapeutic referral for groups or individuals are conceivable, considering personal needs and (rural) living conditions [74,75]. For practitioners, solastalgia as a mediator of psychological disease represents a rather new, but presumably growing, area of work.

This study can provide incentives to improve policy frameworks (e.g., for relocation measures, community participation and consultations), with a stronger focus on the protection and promotion of residents’ and resettlers’ mental health and wellbeing. Since responsibilities are shared between the mining company, municipalities, public authorities and the affected persons themselves, this task is complex. Nonetheless, failing to address it could reinforce power asymmetries and increase the perception of injustice on the part of the local communities.

Following the latest decision of the German government (November 2021) to preserve the old villages at the Garzweiler open-pit mine [89], the mental states of the residents and resettlers might have undergone changes since the study survey. These political developments are likely to be a curse for some and a blessing for others (see Section 3.9). This only reiterates the need for further research and scientific monitoring.

The loss of homes and familiar environments is provoked by both the causes and the consequences of climate change. It is deeply concerning how the collapse of ecosystems can jeopardize mental health on a global and a local level [21,25]. It was only in 2015 that mental health and psychosocial well-being were included in the SDGs, in the context of reducing noncommunicable diseases [90]. Nevertheless, and according to the World Health Organization, multiple goals and targets directly or indirectly support the achievement of good mental health and psychosocial well-being, in particular those SDGs linked to climate hazards, exposure pathways and vulnerabilities [25]. For some study participants, the proximate environmental changes were not the only source of worry but were accompanied by an awareness of the farther-reaching global consequences of burning coal, which could possibly trigger eco-anxiety and ecological grief [6]. The fact that combating the manifold ecological crises can also prevent (mental) illness and promote psychosocial health [21,25] inspires hope. While solastalgia is a place-based experience [7], it is embedded in a global agenda [91]: it exemplifies the potential synergies between the SDGs and recalls that if progress is off-track in one SDG (e.g., climate action, clean energy or life on land), it is likely that others are equally at risk (e.g., good health and well-being), or, put another way: “action on one gets results in the other” [92].

## 5. Conclusions

Mining has displaced millions of people worldwide and created great tensions in local populations and throughout society. The findings of this study suggest that both environmental degradation and (upcoming) resettlement pose risks for depressive, anxious and somatic disorders for local communities in a western German coal-mining region.

Thus, the legitimacy of moving and altering villages for coal should not only be an environmental and climatic debate, but also a public mental health debate. Our data indicate a need to allocate targeted psychosocial support services for affected communities. The links between environmental degradation and mental health are of particular interest to researchers and professionals in the environmental and psychosocial sector, given that this field is still emerging and only partially understood. Furthermore, our findings could have policy implications and could stimulate changes in industry and government decision making and priority setting, to the benefit of residents and resettlers. This study reveals that there are still many unanswered questions. It indicates a pressing need for well-designed, prospective studies that will describe and quantify the (mental) health harms related to coal mining and burning, and the societal costs of these negative externalities.

Albeit drawing upon a German case study, we argue the present findings have international relevance, since large infrastructure projects continue to be implemented today, disrupting natural and human habitats and causing DIDR [93], while the accelerating, man-made climate change will presumably increase solastalgia worldwide.

## Figures and Tables

**Figure 1 ijerph-19-07143-f001:**
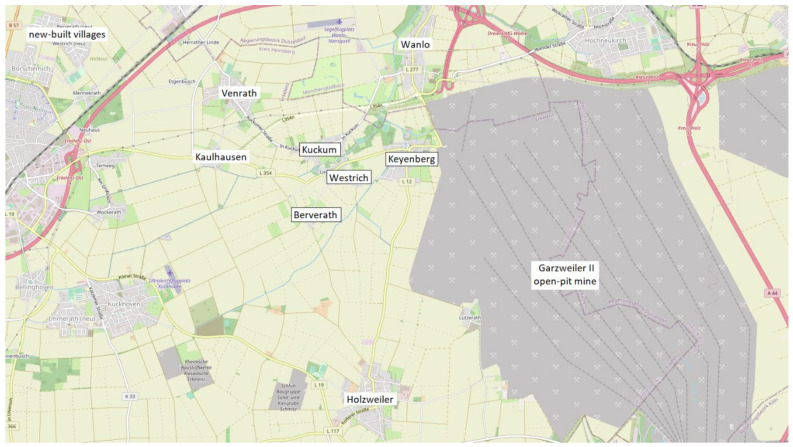
Garzweiler open-pit mine, with relevant surrounding old (framed), pit-edge and new-built villages. © OpenStreetMap contributors (CC BY-SA 2.0), edited by the authors.

**Figure 2 ijerph-19-07143-f002:**
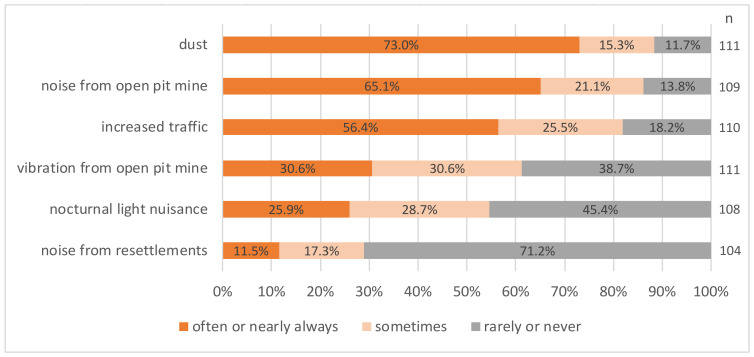
Frequency of observed environmental hazards in old villages.

**Figure 3 ijerph-19-07143-f003:**
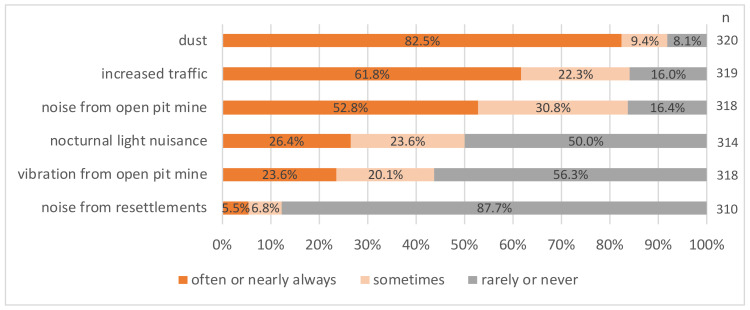
Frequency of observed environmental hazards in pit-edge villages.

**Figure 4 ijerph-19-07143-f004:**
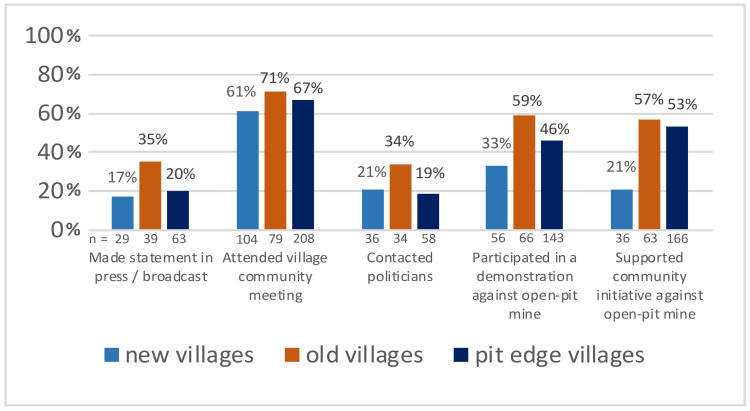
Activities in response to open-pit mining.

**Table 2 ijerph-19-07143-t002:** Sociodemographics, Solastalgia and Patient Health Questionnaire Scores and Correlations.

	New Villages	Old Villages	Pit Edge Villages	*p*-Value *
	Sociodemographics			
	mean (SD)			
age	55.7 (15.7)	54.2 (18.1)	53.9 (15.3)	n.s. ^#^
	*n* = 173	*n* = 104	*n* = 301	
	*n* (%)			
female gender	93 (52.0)	57 (51.4)	177 (55.7)	n.s.
	*n* = 179	*n* = 111	*n* = 318	
marriage or partnership	156 (86.7)	75 (67.6)	261 (81.1)	<0.001
	*n* = 180	*n* = 111	*n* = 322	
university degree	32 (18.7)	26 (25.0)	69 (22.9)	n.s.
	*n* = 171	*n* = 104	*n* = 301	
children living in the village	85 (48.3)	53 (46.9)	130 (40.3)	n.s.
	*n* = 176	*n* = 113	*n* = 323	
grandchildren living in the village	27 (15.3)	9 (8.0)	31 (9.6)	n.s.
	*n* = 176	*n* = 113	*n* = 323	
former generations living in the region	106 (62.0)	81 (73.6)	188 (60.5)	<0.05
	*n* = 171	*n* = 110	*n* = 311	
living on old family property	86 (50.0)	72 (63.7)	131 (41.7)	<0.001
	*n* = 172	*n* = 113	*n* = 314	
ownership of residence	158 (89.8)	98 (89.1)	267 (84.8)	n.s.
	*n* = 176	*n* = 110	*n* = 315	
spend entire life in the village	72 (40.9)	56 (50.0)	117 (37.1)	n.s.
	*n* = 176	*n* = 112	*n* = 315	
	Solastalgia (score)			
	mean (SD)			
	*n* = 170	*n* = 111	*n* = 312	
solastalgia	21.19 (7.51)	25.59 (5.81)	25.38 (4.97)	<0.001 ^#^
- male	20.09 (7.88)	25.85 (5.73)	24.35 (5.93)	<0.001 ^#^
- female	22.34 (6.94)	25.90 (5.27)	26.32 (3.73)	<0.001 ^#^
	Patient Health Questionnaire (score)			
	mean (SD)			
somatization	6.07 (6.71)	10.28 (7.17)	10.05 (7.04)	<0.001 ^#^
- male	6.01 (6.56)	8.69 (6.98)	8.89 (6.86)	<0.05 ^#^
- female	6.09 (6.86)	11.98 (6.84)	11.16 (7.03)	<0.001 ^#^
generalized anxiety	4.60 (5.91)	8.92 (6.07)	7.32 (6.64)	<0.001 ^#^
- male	4.20 (5.24)	7.73 (6.23)	6.22 (5.33)	<0.001 ^#^
- female	4.99 (6.50)	10.19 (5.63)	8.37 (5.70)	<0.001 ^#^
depression	5.02 (6.48)	7.85 (5.86)	7.35 (6.03)	<0.001 ^#^
- male	4.78 (5.98)	7.10 (5.88)	6.50 (6.11)	<0.05 ^#^
- female	5.22 (6.96)	8.70 (5.68)	8.19 (5.87)	<0.001 ^#^
	Patient Health Questionnaire (dichotomized score > 9)			
	*n* (%)			
somatization > 9	47 (28.0)	58 (52.7)	145 (46.5)	<0.001
	*n* = 168	*n* = 110	*n* = 312	
- male	21 (26.3)	24 (46.2)	54 (39.7)	<0.05
	*n* = 80	*n* = 52	*n* = 136	
- female	25 (29.1)	33 (60.0)	88 (52.4)	< 0.001
	*n* = 86	*n* = 55	*n* = 168	
generalized anxiety > 9	31 (18.6)	49 (45.4)	96 (31.2)	<0.001
	*n* = 167	*n* = 108	*n* = 308	
- male	12 (15.2)	17 (33.3)	30 (22.4)	n.s.
	*n* = 79	*n* = 51	*n* = 134	
- female	19 (22.1)	31 (57.4)	65 (39.2)	<0.001
	*n* = 86	*n* = 54	*n* = 166	
depression > 9	35 (20.8)	37 (34.3)	94 (30.3)	<0.05
	*n* = 168	*n* = 108	*n* = 309	
- male	16 (20.0)	17 (33.3)	33 (24.3)	n.s.
	*n* = 80	*n* = 51	*n* = 135	
- female	18 (20.9)	19 (35.2)	60 (36.1)	<0.05
	*n* = 86	*n* = 54	*n* = 166	
	Patient Health Questionnaire and Solastalgia (correlations)			
	r (*n*)			
somatization and solastalgia	0.54 (166)	0.44 (109)	0.42 (310)	<0.001
generalized anxiety and solastalgia	0.51 (165)	0.49 (107)	0.38 (306)	<0.001
depression and solastalgia	0.53 (166)	0.45 (107)	0.35 (307)	<0.001

Respondents from new villages were asked to refer to their village prior to resettlement if necessary; * chi-square or (^#^) Kruskal–Wallis H test; SD = standard deviation; n.s. = not significant; r = Pearson correlation coefficient.

**Table 3 ijerph-19-07143-t003:** Place Attachment and Feelings about Changes caused by Open-Pit Mining.

	New Villages	Old Villages	Pit Edge Villages
	*n* (%) *		
	Place attachment		
I feel a deep connection to this place	68 (39.8%)	79 (73.8%)	228 (74.5%)
	*n* = 171	*n* = 107	*n* = 306
I feel a sense of responsibility to the people of this place	40 (23.7%)	43 (39.8%)	209 (68.3%)
	*n* = 169	*n* = 108	*n* = 306
I feel I have a duty to maintain this place for future generations	28 (16.5%)	60 (56.6%)	242 (79.1%)
	*n* = 170	*n* = 106	*n* = 306
	Positive feelings		
I have understanding for the expansion of the open-pit mine	45 (26.3%)	14 (12.5%)	26 (8.1%)
	*n* = 171	*n* = 112	*n* = 320
Economic benefits of open-pit mining are important for the region	68 (39.5%)	12 (11.0%)	38 (11.9%)
	*n* = 172	*n* = 109	*n* = 320
Funding of community projects by the mining company is helpful for the region	86 (50.9%)	23 (21.1%)	56 (17.7%)
	*n* = 169	*n* = 109	*n* = 317
I am satisfied with efforts of authorities to monitor environmental impacts	47 (27.5%)	10 (9.2%)	24 (7.6%)
	*n* = 171	*n* = 109	*n* = 314
	Negative feelings		
I couldn’t enjoy life as much as I would like to due to the open-pit mine	51 (29.5%)	79 (70.5%)	159 (50.0%)
	*n* = 173	*n* = 112	*n* = 318
My community is/was divided by disagreements over the open-pit mine	101 (58.4%)	76 (68.5%)	121 (38.2%)
	*n* = 173	*n* = 111	*n* = 317
My family is/was divided by disagreements over the open-pit mine	17 (9.8%)	20 (18.4%)	35 (10.9%)
	*n* = 173	*n* = 109	*n* = 320
I am upset at the destruction of historic buildings and landmarks	105 (61.0%)	97 (87.4%)	286 (89.4%)
	*n* = 173	*n* = 111	*n* = 320
I am upset at the destruction of natural habitat for plants and animals	114 (65.9%)	95 (85.6%)	293 (91.6%)
	*n* = 173	*n* = 111	*n* = 320
I am disturbed that future generations are not given higher priority	84 (48.6%)	89 (80.2%)	267 (83.7%)
	*n* = 173	*n* = 111	*n* = 319
I am concerned that my health may be threatened	65 (38.0%)	79 (71.8%)	237 (75.7%)
	*n* = 171	*n* = 110	*n* = 313
I feel powerless against changes of my homeland	102 (60.0%)	89 (79.5%)	267 (85.0%)
	*n* = 170	*n* = 112	*n* = 314

Respondents from new villages were asked to refer to their village prior to resettlement if necessary; * respondents who strongly agree or agree.

**Table 4 ijerph-19-07143-t004:** Experienced or expected Impacts of Resettlement.

	New Villages	Old Villages	*p*-Value ^#^
	*n* (%) *		
feeling physically exhausted	62 (36.3%)	64 (59.3%)	<0.05
	*n* = 171	*n* = 108	
feeling psychologically exhausted	56 (33.0%)	76 (69.7%)	<0.001
	*n* = 170	*n* = 109	
feeling well informed/advised by authorities	55 (32.2%)	9 (8.3%)	<0.001
	*n* = 171	*n* = 109	
(expectation of) better general living conditions	92 (53.8%)	18 (16.7%)	<0.001
	*n* = 171	*n* = 108	
(fear of) lost contact with cherished people	37 (21.6%)	55 (50.9%)	<0.001
	*n* = 171	*n* = 108	
(fear of) extra financial burden	73 (42.7%)	76 (71.0%)	<0.001
	*n* = 171	*n* = 107	
(fear of) worse professional situation	12 (7.0%)	31 (30.4%)	<0.001
	*n* = 170	*n* = 102	
my pets can be kept equally well	58 (73.4%)	23 (36.5%)	<0.05
	*n* = 79	*n* = 63	
my livestock can be kept equally well	4 (28.6%)	1 (3.3%)	<0.001
	*n* = 14	*n* = 30	

* respondents who strongly agree or agree or (for the last two items: pets/livestock) who indicated yes; ^#^ chi-square test.

## Data Availability

Data can be made available upon request.

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
