# Peer review of "A Changing Home: A Cross-Sectional Study on Environmental Degradation, Resettlement and Psychological Distress in a Western German Coal-Mining Region"

_ijerph, 2022, doi:10.3390/ijerph19127143_

Round 1
Reviewer 1 Report
I enjoyed reading this paper and the opportunity to review it. Thank you for your work and for the opportunity. Overall, I think the research is high-quality and makes an important contribution to the field.
I offer several suggested revisions to strengthen the paper.
Abstract:
-
Please review the abstract carefully for grammar and language.
-
Please add a succinct summary of the implications of this work at the end of your abstract.
Introduction:
-
It is a long walk/read to the study aim(s) of your paper. I suggest that you move sub-section 1.3 to Materials and methods under a sub-section called ‘Study context and area’ This will enable you to create a rich description of the place which I think is needed in research on the topic of solastalgia given its place-based nature ( see comments below related to shift in structure/subsection).
Materials and methods:
-
Following the comment above, I suggest re-organizing the structure into the following sub-sections: ‘2.1 Study context and area’ and ‘2.2 Participants and data collection’.
-
I recognize that you note the REB approval in the end matter BUT I do think this should also be included in the text given how important it is. Also, are there any other ethical considerations that should be noted?
-
Please examine and include a brief note as to whether there were important differences between the participants that responded electronically versus paper-based.
-
Please report on how data were entered (paper-based survey), by whom and how data entry errors were avoided and assessed.
-
You note that you extracted and translated “suitable” items into German from the EDS. Please add justification and or process as to how you determined what was suitable.
-
You do not give adequate reasoning for the items used to measure place attachment. There are several existing scales for measuring place attachment (which you are not using) and have also not replicated the items from the EDS. Please provide justification and details for these decisions.
-
Did you pilot test your questionnaire? If not, why not? If not, include this in the limitations section.
-
Briefly justify why you used the PHQ-SADS screening tool (i.e., add one sentence justifying this methodological decision)
-
You do not report any information on response rate, not having this information is a limitation.
Results:
-
For the Sociodemographics sub-section, please include details to let the reader know more information about the representativeness of your study population compared to the population overall (in terms of sociodemographics in particular)
-
I think that the flow of your Results section would be improved if you present the current sub-section 3.4 Env. hazards before the current section 3.3 Solastalgia…
Discussion:
-
Overall, the discussion is well written and you have positioned your findings in relation to existing evidence and research.
-
I would prefer to see your limitations sub-section at the end of the Results section and a more fulsome summary about future research directions in the Discussion section. Currently, there is inadequate synthesis in relation to future directions (which I see as particularly important)
-
The points made on lines 374-381 are very interesting but I think would be better suited later in the discussion section (perhaps in relation to the need for future research given the interest from the communities and participants, in the sub-section I mentioned in the comment above)
-
Return to the question of representativeness of your sample in the limitations section.
References:
- The paper is well-referenced overall.
Author Response
Reviewer 1
Dear Reviewer
Thank you for your helpful comments and suggestions. We have carefully studied them and made our modifications in the text, accordingly. Please find below our detailed response.
Best regards,
Andrea Kaifie, Theresa Krüger and Thomas Kraus
_________________________________________________________________________________
I enjoyed reading this paper and the opportunity to review it. Thank you for your work and for the opportunity. Overall, I think the research is high-quality and makes an important contribution to the field.
Answer: Thank you very much, we really appreciate your valuable feedback and your suggestions that helped to improve our manuscript.
I offer several suggested revisions to strengthen the paper.
Abstract:
Please review the abstract carefully for grammar and language.
Answer: Thank you for this note, we have carefully reviewed the abstract and made corrections in order to improve grammar and language.
Please add a succinct summary of the implications of this work at the end of your abstract.
Answer: Thank you for the suggestion, we added a sentence concerning potential implications, such as “in-depth research, targeted psychosocial support and improved policy frameworks” at the end of the abstract.
Introduction:
It is a long walk/read to the study aim(s) of your paper. I suggest that you move sub-section 1.3 to Materials and methods under a sub-section called ‘Study context and area’ This will enable you to create a rich description of the place which I think is needed in research on the topic of solastalgia given its place-based nature (see comments below related to shift in structure/subsection).
Answer: Thank you for the helpful comment. We restructured the two chapters according to your suggestions and moved most of subsection 1.3 to the subsection 2.1 “Study context and area” in the Material and Methods section. We kept a short paragraph concerning a brief country-wide overview of lignite mining in the Introduction (sub-section 1.3) to create a smooth transition to the study aim (subsection 1.4).
Materials and methods:
Following the comment above, I suggest re-organizing the structure into the following sub-sections: ‘2.1 Study context and area’ and ‘2.2 Participants and data collection’.
Answer: We thank the reviewer for suggesting a more appropriate structuring of this section. As described above, we have moved text from 1.3 to the sub-section 2.1 (Study area and context) and from 2.1 to the sub-section 2.2 (Participants and data collection) to comply with the new structure. We received a competing suggestion from reviewer 2 for the restructuring of the methods section and tried to combine both suggestions. We hope this new structuring is acceptable for the reviewer.
I recognize that you note the REB approval in the end matter BUT I do think this should also be included in the text given how important it is. Also, are there any other ethical considerations that should be noted?
Answer: Thank you for the comment. We have added the REB approval in the new created sub-section 2.5. Ethical considerations. We added a small paragraph concerning psychological distress caused by the questions in the questionnaire.
Please examine and include a brief note as to whether there were important differences between the participants that responded electronically versus paper-based.
Answer: Thank you for this remark. This is an interesting question. We could observe that the population using the online questionnaire was in median younger (49.6 vs. 57.2 years) and more often from villages that were not threatened from resettlement (64.4% vs. 46.4%). We added a sentence concerning the differences in the sub-section 3.1 Sample overview.
Please report on how data were entered (paper-based survey), by whom and how data entry errors were avoided and assessed.
Answer: The data from the paper-based surveys were manually entered into the common database, the SoSci Survey online tool by the author T.Kru. The export of the database was an excel sheet that was screened for errors - first manually for plausibility and then using SAS. The excel table was imported into the SAS program and the descriptive analysis tool for mean, minimum and maximum was used in order to detect entry errors (e.g., significant outliers).
You note that you extracted and translated “suitable” items into German from the EDS. Please add justification and or process as to how you determined what was suitable.
Answer: We thank the reviewers for this remark. Our assessment of the suitability of the items was based on a two-step process. First, we studied the original English-language EDS in detail and made a context analysis through literature research, on-site visits and discussions with key stakeholders and residents. As a result, we considered some items to be not relevant to the situation in the Rhenish mining area. Second, during the piloting of the questionnaire, feedback was obtained from residents on site who rated some items as inappropriate, incomprehensible or too unspecific. Moreover, feedback indicated that the questionnaire was rather too long, as it also contained the three PHQ-SADS scales and detailed questions on resettlement experiences. To create an appropriate balance of length and content, we could not include all items and scales from the EDS.
For example: certain environmental issues (e.g., smoke from household fireplaces, foul smelling or contamination of water) or activities (e.g., made a submission in response to an Environmental Impact Statement, contacted public health or water authorities) did not seem to matter to the residents in the Rhenish mining area according to their assessments. For the solastalgia subscale, two of the original nine EDS items were rated as inadequate or too unspecific (thought of my family being forced to leave this place upsets me; ashamed of the way this area looks now) by key stakeholders in pilot-testing, while two other items were considered as similar (sad that familiar animals and plants are disappearing; unique aspects of nature that made this place special are being lost forever) and therefore converted into a single item (sad that local nature is being damaged).
We have now added two sentences to the procedure in 2.3. for clarification.
You do not give adequate reasoning for the items used to measure place attachment. There are several existing scales for measuring place attachment (which you are not using) and have also not replicated the items from the EDS. Please provide justification and details for these decisions.
Answer: We thank the reviewers for this note. In our paper, we analyzed 3 items on place attachment that were actually replicated from the EDS (deep connection to this place; sense of responsibility to the people of this place; duty to maintain the land/place for future generations). In one item, we preferred the language „place“ instead of „land“ since the word-for-word translation was considered as rather unusual in German language. Other place attachment items from the EDS were not used since they were either considered as inappropriate according to local context, in particular with regard to the complex resettlement situation (e.g., would continue to live in this place even if given the opportunity to leave; would rather live somewhere else; would leave because of changes) or cultural context (e.g. proud of the heritage of this place; feel I know every rock, nook and cranny). Overall, we used only place attachment items that were applicable and appropriate for all three groups, to create certain comparability. In the back translation of the three items the reference to the EDS have may not been obvious anymore, as you indicated. We therefore adjusted the language of the items in Table 3 for clarification.
We refrained from more detailed or metric analysis of the construct place attachment, as this was not the focus of the study. The three EDS items served rather to provide a rougher descriptive assessment of the extent to which place attachment differs or changes between the three groups. The reviewer is completely right, that the application of a reliable intercultural scale on the complex place attachment construct, would have provided better insights and comparisons. We added this additional aspect in the discussion for future research considerations and included a corresponding reference.
Did you pilot test your questionnaire? If not, why not? If not, include this in the limitations section.
Answer: Thank you for this important question. The piloting of the questionnaire was so far only mentioned in the Acknowledgements. We have now added a corresponding sentence on piloting at the end of sub-section 2.3: ”Questionnaires were polit tested by five key stakeholders..”
Briefly justify why you used the PHQ-SADS screening tool (i.e., add one sentence justifying this methodological decision)
Answer: We thank the reviewer for this note and added a brief justification in the subsection 2.3 Survey structure and development, backed up with references.
You do not report any information on response rate, not having this information is a limitation.
Answer: Thank you for this important note. For the online version of the questionnaire, an evaluation of the response rate was not possible as the link was not sent to specific persons. In the paper-based version, 1,900 questionnaires were distributed by drop-off method in person at front doors or in case of absence in mailboxes. 400 questionnaires were placed at two local stores (bakery and gas station). Like the online distribution, an estimation of the response rate would be very imprecise, since no information was available on the number of adults per household (since we were also not able to personally meet people at every front door).
An approximate equivalent of the response rate is shown in Table 1 (% of overall population) for all locations where the drop-off method was used (all old villages, all new villages, Wanlo).
However, we have added this important aspect in the Limitations subsection.
Results:
For the Sociodemographics sub-section, please include details to let the reader know more information about the representativeness of your study population compared to the population overall (in terms of sociodemographics in particular).
Answer: We thank the reviewers for this suggestion. We included details about the representativeness concerning gender and average age for which local data was available (microcensus data). For other sociodemographic items, such as former generations, (grand-)children in the village, comparable data was not available on a population level. Therefore, these items rather served to compare the three study groups among themselves. We added a brief discussion concerning the representativeness of our data in the discussion section.
I think that the flow of your Results section would be improved if you present the current sub-section 3.4 Env. hazards before the current section 3.3 Solastalgia…
Answer: We thank the reviewer for this comment. Since both sub-sections 3.2. and 3.3 refer to table 2, we would prefer to avoid changing the order of the sub-sections. Otherwise, the current chapter 3.4. and the two associated figures 2 and 3 would be located between table 2 and the corresponding sub-section 3.3, which might create confusion. We hope, the reviewer considers this as acceptable.
Discussion:
Overall, the discussion is well written and you have positioned your findings in relation to existing evidence and research.
Answer: We thank the reviewer for the positive evaluation.
I would prefer to see your limitations sub-section at the end of the Results section and a more fulsome summary about future research directions in the Discussion section. Currently, there is inadequate synthesis in relation to future directions (which I see as particularly important).
Answer: We thank the reviewer for the helpful suggestions. We have revised the discussion and transferred parts of the limitations (aspects not explicitly asked for in the questionnaire but frequently mentioned in the comments) to a new sub-section 3.9 Comments of the participants at the end of the results section. We divided the sub-section Limitations and Future directions into two separate sub-sections under discussion. This makes the limitations a little more concise and allows us to elaborate more fulsomely on future research directions, including practical implications of our findings. Here we also added new aspects, based on suggestions from reviewer 2 (SDGs, other pathologies).
The points made on lines 374-381 are very interesting but I think would be better suited later in the discussion section (perhaps in relation to the need for future research given the interest from the communities and participants, in the sub-section I mentioned in the comment above)
Answer: We thank the reviewer for this remark. We moved part of the lines to the new sub-section 3.9. Comments of the participants in the Results section and another part to Limitations (describing why the numbers of returned questionnaires may varied).
Return to the question of representativeness of your sample in the limitations section.
Answer: We thank the reviewer for the comment, which we have added in the 4.1 sub-section Limitations.
References:
The paper is well-referenced overall.
Answer: Thank you very much.
Reviewer 2 Report
I consider that the submitted manuscript meets the requirements to be published in the journal after making some changes to improve it.
ABSTRACT
Review the summary so that it is written in the following format: Introduction, Method, Results and Discussion.
In the abstract some results expressed in % are anticipated and in turn it appears that this difference is significant, the authors must change this aspect, it is not clear to express the results in % and at the same time establish the existence of significant differences (p<.001) , it would be more appropriate to establish this significance with the mean differences obtained (Table 2 and Table 3) reporting the mean and standard deviation instead of the percentage values and some of the values ​​of the Chi-Square tests and Kruskal-Wallis H.
INTRODUCTION
An adequate review of the object of the study is carried out and an analysis is made of the importance of the concept "Solastalgia" (closely related to mining) and of the studies that initially addressed this concept, such as those of Albrecht.
Some effects of solastalgia such as anxiety, stress and attentional fatigue are analyzed, but it would be opportune to include some international studies that analyze its impact on some other pathologies such as obesity, respiratory diseases, attention deficit disorder, among others, both in the introduction as in the discussion of the results.
The introduction is adequate and up to date but could be further improved and updated.
In this sense, this reviewer has not found references or relationships (which are clear) with the 17 sustainable development goals of the United Nations (2022), it would be appropriate to establish an analysis of the aspects most linked to this study (introduction, conclusions, etc. .).
MATERIALS AND METHODS
It is suggested to change the name of the sub-sections of the materials and methods section that have been considered and call them: participants, instruments, procedure and data analysis (Chi-Square tests and Kruskal-Wallis H, etc.), following the guidelines of some international regulations (such as the APA standards, since they are mentioned in the study in the discussion and conclusions).
When justifying the analyzes carried out, it would be opportune to indicate that the tests carried out are non-parametric, probably because the distribution of the data is not normal.
RESULTS
They clearly present the evidence obtained in the study, but the authors should explicitly report the results of the Chi-Square tests and Kruskal-Wallis H tests, following the recommendations for preparing research reports.
DISCUSSION-CONCLUSIONS
It is adequate, but it is recommended to include the suggestions indicated in the introduction
REFERENCES
The index of topical news is high, the vast majority of sources are from recent years.
Author Response
Reviewer 2
Dear Reviewer
Thank you for your helpful comments and suggestions. We have carefully studied them and made our modifications in the text, accordingly. Please find below our detailed response.
Best regards,
Andrea Kaifie, Theresa Krüger and Thomas Kraus
_________________________________________________________________________________
I consider that the submitted manuscript meets the requirements to be published in the journal after making some changes to improve it.
Answer: We thank the reviewer for the positive evaluation and helpful comments that further improved the quality of the manuscript.
ABSTRACT
Review the summary so that it is written in the following format: Introduction, Method, Results and Discussion.
Answer: We thank the reviewer for this note. We agree with the author, that a subsection design in the Abstract is helpful in order to understand the study aim, results and conclusion. Unfortunately, the Abstract requirements for IJERPH dictate a single paragraph Abstract without headings.
In the abstract some results expressed in % are anticipated and in turn it appears that this difference is significant, the authors must change this aspect, it is not clear to express the results in % and at the same time establish the existence of significant differences (p<.001) , it would be more appropriate to establish this significance with the mean differences obtained (Table 2 and Table 3) reporting the mean and standard deviation instead of the percentage values and some of the values ​​of the Chi-Square tests and Kruskal-Wallis H.
Answer: We thank the reviewer for this remark. In our study, we estimated the mean mental health scores for depression, anxiety as well as somatization. In addition, we dichotomized the score to detect the proportion of people with a at least moderate symptom burden. In accordance with the reviewer, we not only reported the proportion of at least moderate symptoms but also added, exemplarily for depression, the mean score values with SD for self-reported depressive symptoms including p-test result.
INTRODUCTION
An adequate review of the object of the study is carried out and an analysis is made of the importance of the concept "Solastalgia" (closely related to mining) and of the studies that initially addressed this concept, such as those of Albrecht.
Answer: We thank the reviewer for the positive evaluation.
Some effects of solastalgia such as anxiety, stress and attentional fatigue are analyzed, but it would be opportune to include some international studies that analyze its impact on some other pathologies such as obesity, respiratory diseases, attention deficit disorder, among others, both in the introduction as in the discussion of the results.
Answer: We thank the reviewer for this helpful suggestion. We addressed the relationship between solastalgia and other pathologies in the introduction in sub-section 1.1 The concept of solastalgia including references, as well as in section 4 Discussion in the context of perceived health threats.
The introduction is adequate and up to date but could be further improved and updated.
In this sense, this reviewer has not found references or relationships (which are clear) with the 17 sustainable development goals of the United Nations (2022), it would be appropriate to establish an analysis of the aspects most linked to this study (introduction, conclusions, etc.).
Answer: We thank the reviewer for this very important aspect. We have now embedded the connection of solastalgia with the SDGs (3, 7, 13, 15) / 2030 Agenda in both introduction (end of subsection 1.1) and discussion, including relevant and topical references.
MATERIALS AND METHODS
It is suggested to change the name of the sub-sections of the materials and methods section that have been considered and call them: participants, instruments, procedure and data analysis (Chi-Square tests and Kruskal-Wallis H, etc.), following the guidelines of some international regulations (such as the APA standards, since they are mentioned in the study in the discussion and conclusions).
Answer: Thank you for this remark. We received a competing suggestion from reviewer 1 for the restructuring of the methods section and tried to combine both suggestions. We hope this new structure is acceptable for the reviewer.
When justifying the analyzes carried out, it would be opportune to indicate that the tests carried out are non-parametric, probably because the distribution of the data is not normal.
Answer: We thank the reviewer for this remark and have added a corresponding sentence for clarification in sub-section 2.4 Analysis.
RESULTS
They clearly present the evidence obtained in the study, but the authors should explicitly report the results of the Chi-Square tests and Kruskal-Wallis H tests, following the recommendations for preparing research reports.
Answer: We thank the reviewer for this suggestion and included the exact results of the Chi-Square and Kruskal-Wallis H tests (Table 2 and 4) in the suppl. (Table S3 and S4), with corresponding references in the main paper. To ensure consistency, we have also edited suppl. Table S2 (exact p-values, instead of n.s. = not significant).
DISCUSSION-CONCLUSIONS
It is adequate, but it is recommended to include the suggestions indicated in the introduction
Answer: We thank the reviewer for this remark. We added a small paragraph regarding pathologies in the Discussion (Line 520) and a small paragraph concerning the SDGs at the end of the sub-section 4.1. Implications and future directions.
REFERENCES
The index of topical news is high, the vast majority of sources are from recent years.
Answer: We thank the reviewer.